# Why are patients with acute traumatic brain injury not routinely assessed or treated for vestibular dysfunction in the UK? A qualitative study

Rebecca M Smith  ,[1] Caroline Burgess,[2] Vassilios Tahtis,[3] Jonathan Marsden,[4] Barry M Seemungal  [1]

¹Brain and Vestibular Group, Centre for Vestibular Neurology, Imperial College London, London, UK
²School of Population Health & Environmental Sciences, King's College London, London, UK
³Therapy Department, King's College Hospital, London, UK
⁴School of Health Professions, Plymouth University, Plymouth, UK

**Correspondence to**
Dr Barry M Seemungal; b.seemungal@imperial.ac.uk and Ms Rebecca M Smith; rebecca.smith@imperial.ac.uk

## ABSTRACT

**Objectives** Vestibular dysfunction is common in patients with acute traumatic brain injury (aTBI). Persisting vestibular symptoms (ie, dizziness and imbalance) are linked to poor physical, psychological and socioeconomic outcomes. However, routine management of vestibular dysfunction in aTBI is not always standard practice. We aimed to identify and explore any healthcare professional barriers or facilitators to managing vestibular dysfunction in aTBI.

**Design** A qualitative approach was used. Data were collected using face to face, semi-structured interviews and analysed using the Framework approach.

**Setting** Two major trauma centres in London, UK.

**Participants** 28 healthcare professionals participated: 11 occupational therapists, 8 physiotherapists and 9 surgical/trauma doctors.

**Results** Vestibular assessment and treatment were not routinely undertaken by trauma ward staff. Uncertainty regarding responsibility for vestibular management on the trauma ward was perceived to lead to gaps in patient care. Interestingly, the term dizziness was sometimes perceived as an 'invisible' and vague phenomenon, leading to difficulties identifying or 'proving' dizziness and a tendency for making non-specific diagnoses. Barriers to routine assessment and treatment included limited knowledge and skills, a lack of local or national guidelines, insufficient training and concerns regarding the practical aspects of managing vestibular dysfunction. Of current trauma ward staff, therapists were identified as appropriate healthcare professionals to adopt new behaviours regarding management of a common form of vestibular dysfunction (benign paroxysmal positional vertigo). Strategies to support this behaviour change include heightened clarity around role, implementation of local or national guidelines, improved access to training and multidisciplinary support from experts in vestibular dysfunction.

**Conclusions** This study has highlighted that role and knowledge barriers exist to multidisciplinary management of vestibular dysfunction in aTBI. Trauma ward therapists were identified as the most appropriate healthcare professionals to adopt new

## STRENGTHS AND LIMITATIONS OF THE STUDY

⇒ This multicentre qualitative study is the first to explore barriers and facilitators to managing vestibular dysfunction in acute traumatic brain injury (aTBI) in a range of healthcare professionals.
⇒ Data were analysed using a systematic, transparent approach, framework analysis, to heighten rigour and trustworthiness within the results.
⇒ While the study size was modest, healthcare professionals were purposefully sampled to encompass a range of professional backgrounds and experience.
⇒ Although patients were recruited from two sites, this represents half of the major trauma centres in London, UK, supporting the transferability of findings to other urban major trauma centres in the UK.
⇒ This study investigates healthcare professionals' barriers and facilitators to managing vestibular dysfunction in aTBI, future work could usefully explore patients' and carers' experiences.

behaviours. Several strategies are proposed to facilitate such behaviour change.
**Trial registration number** ISRCTN91943864.

## INTRODUCTION

Vestibular dysfunction in traumatic brain injury (TBI) linked to injury of peripheral (ie, inner ear and nerve) or central (ie, brain) vestibular structures can result in dizziness or imbalance[1] and is very common, affecting up to 80% of ambulant moderate-to-severe patients with acute TBI (aTBI).[2 3] Vestibular dysfunction in aTBI may be caused by a range of diagnoses including benign paroxysmal positional vertigo (BPPV), centrally mediated gait ataxia (typically a 'vestibular ataxia'), migraine phenotype headache and acute peripheral unilateral vestibular loss.[4] Patients typically present with multiple vestibular diagnoses,[4 5] increasing the complexity of assessment and treatment and elevating the risk of missed diagnoses.

Early management of vestibular dysfunction following TBI appears to be important. Delays to or inaccurate diagnosis and treatment may adversely impact patients' physical and psychosocial outcomes and quality of life,[6–8] while persisting vestibular symptoms have been found to delay return to work.[9] Vestibular dysfunction, by its link to falls[10–12] (which affect half of TBI survivors[13]), results in significant physical, psychological and healthcare costs.[14–16] Evidence also points to the direct impact of vestibular dysfunction on mental well-being, with studies reporting links between the vestibular system and brain areas involved in emotional and cognitive processing.[17–19] Despite the need for early intervention, routine vestibular assessment and treatment during aTBI does not appear to be commonplace. Previous studies noted long delays to diagnosis and treatment,[8] and large discrepancies in assessment and treatment practices between trauma centres.[20] Potential explanations for a lack of routine vestibular management in aTBI include: (1) limited ward-based vestibular expertise, (2) absence of recommendations for vestibular assessment or treatment in national early management of head injury guidelines[21] and (3) a newly described but common clinical phenomenon called vestibular agnosia (blunting of vestibular perception in patients with moderate-severe aTBI).[22]

The lack of vestibular expertise on major trauma wards is perhaps not surprising since the complexity of vestibular dysfunction has only recently been elucidated,[4 5 22] and vestibular neurologists/specialists do not appear in the current list of recommended trauma ward-based healthcare professionals.[23] Whether the latter is a consequence of or contributing factor to national guidelines not stipulating the need of routine vestibular assessment or management is unclear, however the two factors may well be interdependent. Thus, major trauma wards are often staffed by surgical specialities who are expert at acute life-saving interventions but less so at managing the neurological complications of TBI (dizziness, imbalance and cognitive sequelae). Perhaps challenging for healthcare professionals is the finding that vestibular dysfunction in aTBI, with its attendant increased risk of falls, can be 'silent' because of disrupted vestibular perception causing a vestibular agnosia (linked to a sevenfold reduction in recognition of common diagnoses such as BPPV by ward staff).[22] Traditional teaching for healthcare professionals is to perform a focused examination based on the history. However, in aTBI, the more significant the brain injury the higher the underlying vestibular burden, but the less likely the patient is to report symptoms. For example, BPPV is twice as common in aTBI with skull fracture than in patients without skull fracture; that is 33% vs 66%.[24] The combination of a very high vestibular burden in aTBI, its silent nature and the standard approach to perform symptom-specific examination results in many patients being discharged home without any vestibular diagnoses, let alone specific treatment.

In summary, although early vestibular assessment and treatment in all patients with aTBI is warranted, it is not routinely implemented. Prior to any change in policy or practice, it is imperative to understand the views of those routinely delivering the service or those who are likely to be affected by it.[25] Accordingly, we aimed to explore healthcare professional barriers or facilitators associated with screening, assessment and treatment of vestibular dysfunction in patients with aTBI .

## MATERIALS AND METHODS
### Patient and public involvement
Patients were involved in aiding identification and prioritisation of the research question and study design. More specifically, during an information session, patients from a local TBI association reported suboptimal experiences of vestibular dysfunction management in aTBI, while healthcare professionals were perceived to poorly understand symptoms of vestibular dysfunction. Such feedback formed the basis of the present study.

### Study design
A qualitative methodology, the Framework approach, was used to gather experiential data.[26 27] Originally this methodology was developed for large-scale policy research, but has been utilised more widely in health research[28] and pertinently, in studies exploring barriers to diagnostic and treatment implementation.[29–32]

### Sampling, sample size and participants
Clinicians were invited by email to participate if (1) they routinely worked on a trauma, emergency, rehabilitation or other ward receiving patients directly from acute trauma wards, and (2) had a role in routine assessment and treatment of patients with aTBI . Purposive sampling was used to obtain a sample of healthcare professionals with differing levels of experience. Guidance from previous studies using Framework analysis and discussions regarding data saturation were used to determine an appropriate sample size.[33 34] This sampling method is in line with similar studies.[35 36]

### Data collection and setting
Semi-structured, individual, face-to-face interviews were conducted by the same researcher (RS), using a topic guide (online supplemental file 1) to gather in-depth data.[37–39] Interviews were audio-recorded and transcribed verbatim. Written consent was obtained from all participants.

The theoretical domains framework (TDF) was utilised to inform the topic guide. The TDF was used to (1) allow a greater understanding of factors influencing clinical behaviour, (2) determine possible strategies to change behaviour and (3) clarify how such strategies might be best executed.[40] The TDF was developed to identify psychological and organisational theory relevant to health practitioner behaviour change; culminating in 12 domains covering factors such as knowledge, skills and social and professional roles.[41] The TDF may therefore provide a

theoretical model for any subsequent behaviour change intervention, and further, may enable successful implementation of that intervention. The topic guide included questions on how symptoms of vestibular dysfunction that is, dizziness and imbalance, were managed on the trauma ward. Prompts and probes were utilised to stimulate further discussion, if required.[42] This approach has also been used in other studies employing the TDF and Framework analysis.[33 43] Participants were not asked to define dizziness or imbalance as this did not form part of the primary research question.

### Data analysis

Data were analysed utilising the Framework approach; a series of interconnected stages enabling the researcher to move back and forth across the data until a coherent account emerges. During analysis the data are charted, and sorted according to key themes.[44] Two researchers reviewed and refined the framework (online supplemental file 2), which underwent several iterations before it encompassed the whole dataset. Using NVivo (V.12), the data were charted, whereby indexed data were summarised for each participant. After charting 10 transcripts, data were discussed among the research team. Some subthemes were noted to be redundant or hold significant overlap and were removed or renamed. Following refinement of the framework, the final transcripts were charted.

Mindmaps (MindView V.7) were created for each of the five themes, providing a visual representation of the whole dataset. Notes were made on connections, patterns and areas of convergence and divergence between participants, subthemes or themes. Finally, a central chart (online supplemental file 3) was created encompassing all respondents across both sites. This was used to explore patterns across themes and participant groups.

### RESULTS

Results are reported in accordance with Consolidated Criteria for Reporting Qualitative Research guidelines where possible.[45] Thirty-five healthcare professionals across two Major Trauma Centres were invited to take part: of those, seven declined to take part or did not reply to the invitation email and 28 participated. Those declining to take part were evenly spread regarding their profession and level of experience. Table 1 illustrates the demographics of the 28 participants. In our sample, healthcare professionals had on average 47.4 months experience of working in trauma.

Five main themes relating to healthcare professionals' experiences of managing vestibular dysfunction are outlined: (1) Current practice—who is responsible for screening, assessing and treating vestibular dysfunction? (2) The invisibility of dizziness: how clinically important is it? (3) How confident are healthcare professionals in their knowledge and skills to assess and treat vestibular dysfunction? (4) What are the practical barriers to assessment and treatment? and (5) Who and what is required for behaviour change? These five main factors were noted to be connected by an overarching characteristic: Healthcare professionals' role (figure 1). Quotes from participants are included to illustrate each of the five main themes. Quotes are followed by the pseudonym, profession, and specialty of the participant.

### Current practice—who is responsible for screening, assessment and treatment of vestibular dysfunction?

Trauma and surgical doctors across both sites felt a theoretical sense of oversight for all trauma-related deficits, however there was uncertainty regarding responsibility for day-to-day management of vestibular dysfunction. Doctors appeared less likely to ask about dizziness or assess vestibular dysfunction routinely. However, one divergent case, a consultant neurosurgeon, reported screening patients more frequently although this was guided by the presence of skull fractures on a scan, rather than a routine question.

> No, we absolutely don't assess it routinely… it's not something that we directly asked questions about, so we often missed it.
>
> (A13, Registrar, Surgery)

Therapists (physiotherapists and occupational therapists) also expressed uncertainty about responsibility for managing vestibular dysfunction, although in practice they tended to identify patients with vestibular dysfunction and coordinate referrals. Therapists attributed taking on these roles due to spending more time with patients

**Table 1** Demographics of study participants

| Healthcare professional | Number interviewed | Female | Specialty (number) |
|---|---|---|---|
| Junior therapist | 7 | 5 | Physiotherapist (4) Occupational therapist (3) |
| Senior therapist | 12 | 10 | Physiotherapist (4) Occupational therapist (6) |
| Junior doctor | 5 | 2 | Surgery (3) Trauma (2) |
| Senior doctor (registrar–consultant) | 4 | 1 | Neurosurgery (2) Trauma (2) |

# Role

| 1 - Responsibilty | 2 - Clinical need | 3 - Knowledge and Skills | 4 - Practical barriers | 5 - Behaviour change |
|---|---|---|---|---|
| Role affects perception of responsibility for vestibular management | Role influences perception of clinical importance of vestibular dysfunction | Role impacts confidence in specific knowledge and skills | Role influences practical barriers to managing vestibular dysfunction | Role acts as a facilitor or barrier for behaviour change |

**Figure 1** Five main themes relating to healthcare professionals' experiences of managing vestibular dysfunction and their relation to the overarching concept of role.

and being the first healthcare professional to mobilise or complete functional tasks with patients (these activities often provoked dizziness). Large variability was noted in how therapists identified patients; some routinely asked about dizziness or imbalance, whereas others relied on patient report or manifestations of vestibular dysfunction in patients' balance, gait or body language. Following identification, therapists referred patients to specialists. No therapists at either site reported completing specific assessments or treatments independently. There was a sense among occupational therapists in particular, that managing vestibular dysfunction was outside of their remit and instead embedded in a physiotherapist's role (due to their existing involvement in balance assessment).

> I think when you're an OT, probably the expectations were that you're not the one doing that. I would have no problem to do it if I felt that I was trained to do it. I think it's just never been something I've ever been encouraged to look into because, generally, the physios come along and do it…I think it might be a cultural thing within therapy.
>
> (A6, Occupational therapist, Senior Specialist)

Respondents reported variability concerning when and to whom patients were referred. At both sites, specialists were visiting teams and therefore not regularly ward or sometimes even site based, leading to variation in time-to-assessment and occasionally patients being discharged prior to assessment. Interestingly, although specialist input was universally noted to be positive, respondents also viewed the presence of specialists as a barrier to improving their own vestibular knowledge and skills. Uncertainty was also evident in which patients were eligible for follow-up post discharge. Therapists appeared to devolve responsibility to patients to contact their general practitioner if they had ongoing symptoms of vestibular dysfunction. One participant described this could lead to patients 'falling through gaps'. The uncertainty around responsibility and the variability noted in treatment pathways were thought to have a negative impact on vestibular care during and following admission.

> I don't know about a clear pathway, there's not really a protocol I should say, it's bit more adhoc…The service is not really equitable for everybody, and also probably people are lost to follow up and have poor outcomes in the future.
>
> (A5, Physiotherapist, Senior)

## The invisibility of dizziness: how clinically important is it?

There were marked differences between doctors and therapists regarding their perception of the clinical importance of dizziness, seemingly related to expectations of their role. Doctors perceived dizziness as a short-lived symptom and where it did persist, it could be managed in outpatient rather than inpatient settings.

> Not a massive priority…I think the feeling is that if someone has a head injury, they probably have a bit of, they could well have some dizziness, but it may not you know, it'd probably just be transient.
>
> (A27, Registrar, Trauma)

Doctors' views on the importance of dizziness appeared to be related to their clinical priorities of immediate life preservation or signs of acute deterioration. As one participant described 'we often take our foot off the pedal a little bit and dismiss other things as unimportant'. This seemingly manifested in how frequently they asked about vestibular symptoms. Doctors noted the impact of dizziness on patients' confidence and balance, but generally were more ambivalent about it causing direct harm. Notably there was the feeling that it could delay discharge.

> These patients are not so bad that they have to be seen within 24 hours, even if you delayed it [treatment] by two or three days it's not a big issue. It increases their hospital stay but it doesn't cause any harm to them, so it's not a big problem.
>
> (A8, Consultant, Neurosurgeon)

Interestingly, 'subtle' or 'invisible' were words used to describe dizziness, attributed to ward round assessments being conducted while patients were lying still when signs of vestibular dysfunction were not always apparent (to a

non-expert) and/or dizziness was not reported. Further, respondents noted the subtlety, subjective and positional nature of dizziness was not only a barrier to identification (and therefore accurate assessment and treatment), but also in 'proving' patients were dizzy, which was further seen to limit its clinical importance. Respondents did not mention the potential for objective measurement of vestibular dysfunction and hence were likely unaware of the capability for definitive diagnosis via laboratory testing.

> It's not seen on a scan and so often it's the importance of it is hard to emphasise to the wider sort of medical community so whereas if it's a blood result you can show it and I think the impact is probably underestimated. As physiotherapists we probably know the impact of it but generally, I'm not convinced the wider medical community recognises the repercussions.
>
> (A3, Physiotherapist, Senior)

Therapists viewed dizziness as a higher priority; increasing risk of falls, impacting cognition, attention, confidence, independence and emotional and social well-being. Additionally, dizziness was noted to impede progress with recovery, resulting in an increased length of stay, heightened demand on ward therapy staff and more support at home.

> If they're feeling dizzy, they spend longer periods in bed and they're up and walking around less, which then obviously has a lot of other secondary complications in terms of prolonged bed rest and not moving around and decreased appetite or decreased oral intake, just because they're struggling to get up. And then I think it can increase their length of stay.
>
> (A11, Physiotherapist, Senior Specialist)

### How confident are healthcare professionals in their knowledge and skills to assess and treat vestibular dysfunction?

Across all respondents, there was some theoretical knowledge but limited ability or confidence with practical vestibular assessment and treatment skills. Therapists denied knowledge or use of vestibular assessment tools, while there was wider awareness and use of 'balance measures' and assessment for postural hypotension. Low confidence was noted in undertaking eye movement examinations, and interpretation of findings was felt to be out of the scope of their knowledge and skills.

> I don't routinely do an actual dizziness assessment. I'd maybe look into balance and see whether they've got poor balance which might be linking everything in… I don't have a specific assessment to do.
>
> (A10, Physiotherapist, Junior)

Respondents exhibited some theoretical knowledge of how to undertake clinical bedside diagnostic and treatment manoeuvres for the most common cause of peripheral vestibular dysfunction (BPPV), but little or

no confidence in practical skills. Where there was practical experience, this was limited to physiotherapists who were not routinely treating patients due to (1) low confidence secondary to limited patient exposure and insufficient training and mentorship, (2) a reliance on visiting specialists or (3) the practicality of undertaking treatment in patients with aTBI. Across both sites therapists had little confidence in other therapy or medical staff to manage BPPV. Therapists noted specialists were effective at treating BPPV, although there was divergence regarding dosing and optimum time to treat.

> I don't think any of us up here feel confident to do it…I also think because the [trauma] doctors are rotational and don't necessarily have in-depth knowledge, they won't be confident to prescribe or treat. They definitely don't know how to do the manoeuvres.
>
> (A5, Physiotherapist, Senior)

Similarly, trauma or surgical doctors did not report use of specific vestibular assessments. There was more confidence in completing general eye movement examinations, although interpretation of findings and discerning peripheral (ie, inner ear) versus central (ie, brain) patterns was felt to be complex. Some theoretical knowledge of the BPPV diagnostic test was apparent, but there were lower levels of confidence in their (or their therapy or medical colleagues) ability to practically undertake bedside tests or treatment. Non-specific treatments such as medication to suppress vertigo were most frequently reported as first-line treatment.

> I don't think anyone in my team including myself would confidently say we can deal with it…I don't think the therapists would go to the extent of actually doing specific manoeuvres, the Hallpike manoeuvre.
>
> (A8, Consultant, Neurosurgeon)

### What are the practical barriers to assessment and treatment of vestibular dysfunction?

Participants perceived training and knowledge as the most fundamental barrier to managing vestibular dysfunction. Secondary barriers were both intrinsic (motivation and confidence) and extrinsic (time and the feasibility of completing diagnostic and treatment procedures). While both physiotherapists and occupational therapists felt their previous vestibular training did not enable them to assess and treat independently, occupational therapists perceived their training was not comparable to that of physiotherapists. Although neither physiotherapists nor occupational therapists felt managing vestibular dysfunction was an expectation of their role, occupational therapists felt concerned that taking an interest would cross professional boundaries. Dizziness and imbalance were still perceived to be relevant to their role, however.

> I'm not sure whether we would be um sort of stepping in some area that is not supposed to be ours however in terms of occupational therapy, it's something that

affects function so I think it's relevant…we should be more aware of how to treat.

(A4, Occupational therapist, Junior)

For those receiving more training, limited or adhoc exposure to patients and hence reduced application of practical skills, coupled with the rotational nature of training seemingly affected confidence, knowledge, and skill consolidation.

I haven't actually been taught how to do it [assessment and treatment of BPPV], I've just been shown it, or told the basis behind it, rather than actually being taught to carry it out…I think rotating out of it all the time and just not, never really having the chance to consolidate skills… again it's not really something that we're expected to manage.

(A11, Physiotherapist, Senior Specialist)

Doctors recalled some undergraduate teaching but felt this was not revisited during clinical training, perhaps associated with their focus on managing acute aspects of TBI. Further, there was a perception among doctors at both sites of dizziness as an ill-defined symptom, without discrete cause or diagnosis and without specific treatment. This view of dizziness as 'unfixable' seemed particularly important for neurosurgeons who were accustomed to being able to 'fix' symptoms. Interestingly, this perception was not noted among therapists.

There is this concept of post-concussion dizziness where it's really non-specific and the idea is it's not really treatable, it's not a specific condition. And so if it's not treatable and particularly as surgeons our mindset is to only think about things you can fix…I think internationally that post-concussion dizziness isn't discrete diseases, it's just a fluffy phenomenon that occurs as a sequelae of head injury.

(A13, Registrar, Surgery)

Time constraints were particularly evident among doctors and therapists working in emergency areas, seemingly due to competing priorities. Contrastingly, trauma therapists highlighted two secondary barriers (1) paucity of role models and (2) feasibility of identifying vestibular dysfunction and performing assessments and treatments in patients with aTBI due to spine and limb fractures, pain, consciousness, communication, cognition and insight.

### Who and what is required for behaviour change?
When asked about incorporating managing vestibular dysfunction into their role, trauma therapists displayed the most enthusiasm about behaviour change. This was corroborated by doctors, who perceived therapists to have adequate time and to see patients at an appropriate point in their recovery. Trauma therapists were specific about exclusively managing BPPV, rather than other causes of vestibular dysfunction.

If I had the right training, I'd be very happy to go and treat, considering that it's [BPPV treatment] almost 100% effective and its quicker turnaround, absolutely. I don't think, why would we need to waste more resources and time really? Because what am I just screening in order to get someone else to come and fix the problem? So, it might be better if we were trained, I think it would be in our scope.

(A25, Occupational therapist, Senior Specialist)

Reservations to behaviour change included (1) concerns regarding remit (occupational therapists) and (2) concerns around staff capacity and transferability of skills (physiotherapists). Facilitators involved changes to local or national guidelines and accessible information for healthcare professionals' and patients to help consolidate awareness and knowledge.

Some sort of policy or practice because I don't think it's enough just to do some training. Because I think people go for these trainings, then just don't do it. I think has to maybe come from something bigger, like a change in policy….or something locally like a policy on head injury management or vestibular management.

(A7, Physiotherapist, Team Lead)

Role-related facilitators included setting clear expectations, regular patient exposure, vestibular neurology team support for complex cases, training and mentorship and endorsement from line managers. Heightened theoretical and practical training and training a range of therapists to ensure sufficient capacity and maintenance of skills was felt necessary. A theoretical and practical checklist was thought to improve confidence and ensure consistency. Potential benefits to behaviour change included more timely assessment and treatment, shortened hospital stay, improved progress with therapy and fewer patients with BPPV being missed.

I think it would only benefit the patient because actually earlier on we'd be focusing on all aspects and picking it up better… And I think as well if you can settle someone's dizziness earlier on, they'll engage better with the therapy along the way because actually if they're always dizzy and we're struggling with that for a while and we're waiting for that assessment and then the treatment, you know you want the patient to build confidence.

(A26, Physiotherapist, Senior Specialist)

## DISCUSSION
### Summary of findings
This study provides new insights into the impact of healthcare professional's perceived role on vestibular assessment and treatment behaviours in aTBI. Our findings suggest management of vestibular dysfunction may be affected by (1) uncertainty within healthcare professionals' role

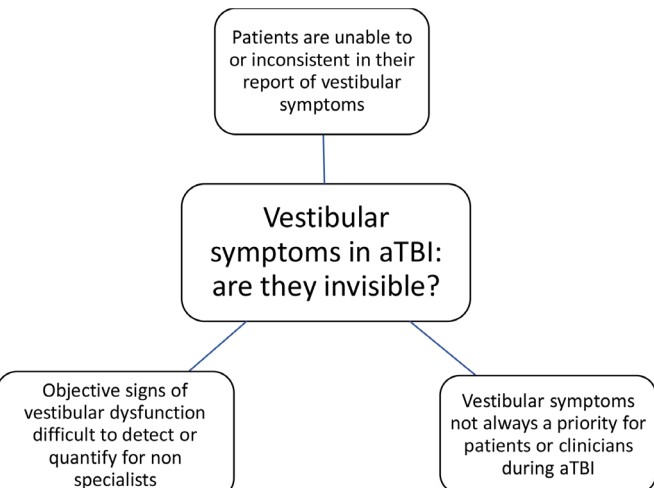

**Figure 2** Diagram noting the themes contributing to the 'invisibility' of vestibular dysfunction. aTBI, acute traumatic brain injury.

and assessment and treatment pathways, (2) self-reported lack of confidence in practical skills, (3) lack of training, access to a multidisciplinary team for complex cases and mentorship and (4) a perception of dizziness as an invisible and unfixable entity. Optimism was expressed however, towards the possibility of behaviour and role change to improve the care pathway.

The role-related uncertainty displayed by healthcare professionals may be linked to lack of detail in existing guidelines in how, when and by whom post-traumatic vestibular dysfunction should be managed.[21 23] Previous research notes sufficient clarity surrounding role is significant in relation to completing and focusing on important tasks,[46] while implementation of guidelines in other areas of aTBI management noted improved practice and cost savings.[47] Thus, heightened clarity around clinicians' roles and formation of a care pathway or guidelines may be useful in improving care for patients with aTBI with vestibular dysfunction.

Insufficient training and a scarcity of role models were also perceived to be barriers to routine care. The degree to which vestibular management is included in undergraduate medical or therapy training is unclear, however published surveys note training is variable and appears to be undertaken at postgraduate level.[48 49] To date there are no formal qualifications for therapists involved in managing general vestibular dysfunction although draft proposals are in process for physiotherapists.[50] Such subspeciality training could involve integration with a larger infrastructure providing access to expert review and audit of complex cases, training, research opportunities and mentorship. The lack of available role models noted by our participants is also noteworthy given the documented benefits of clinical behaviour, identity and career development.[51 52] Indeed, the absence of role models may have contributed to the ambiguity around responsibility for patient care and thus would be important to address for behaviour change.

Our participants described the term dizziness as an invisible or subtle phenomenon. Previous research corroborates this, noting (1) healthcare professionals perceive dizziness to be a vague symptom[53]; (2) patients themselves exhibit inconsistencies in subjective reporting of dizziness[54] and (3) when patients were asked about their perceptions of living with dizziness, 'invisible' was commonplace.[55 56] Importantly, these latter studies noted patients with chronic dizziness associated invisibility with a lack of self-validation and validation from healthcare professionals, thus giving weight to the rationale for early diagnosis and treatment in patients with aTBI. Dizziness was described as 'subtle' by participants in our study, despite objective signs of vestibular dysfunction being elicitable (although by experts) in the majority of patients with aTBI.[4 24] Given many patients complain of few or no vestibular symptoms (due to vestibular agnosia),[22] it is perhaps not surprising that patients' with aTBI perceived dizziness is indeed, subtle (figure 2). It follows that trauma and surgical doctors view of dizziness as a non-specific entity, without need for further diagnosis or specific treatment is understandable, although incorrect given the latest research.[4 22 24] Indeed, the general term 'post-traumatic dizziness' has historically been used in literature and practice rather than specific diagnoses, although recent work highlights the need to diagnose discrete conditions.[4 5 57] A specific diagnosis is important for accurate treatment and for self-validation,[56 58] which in turn can influence attitudes and beliefs.[59] Further high-quality treatment studies may encourage the use of specific vestibular diagnoses among healthcare professionals, thus providing patients with accurate treatment and validation of their symptoms.

Surgical and trauma ward doctors assumed oversight for trauma patient care, however due to their focus on life-threatening complications of aTBI and the specialist nature of managing vestibular dysfunction, they tended to feel managing dizziness and imbalance were better suited to visiting specialists. Therefore, it is perhaps not practical to expect surgical or trauma doctors to acquire the necessary competency to manage vestibular dysfunction, and thus new models of care should incorporate clinicians who either already were sufficiently trained or had capacity to be trained and mentored. From our sample trauma therapists (physiotherapists and occupational therapists) felt most able to adopt new assessment and treatment behaviours, limited to managing BPPV rather than all causes of post-traumatic vestibular dysfunction. This is unsurprising as vestibular presentations may interact with other complex post-traumatic neurological conditions such as epilepsy, where some treatments may worsen dizziness and imbalance and delay discharge,[60] or vestibular migraine the most common acute manifestation of which is gait ataxia.[61] An optimal scenario may involve a team; therapists trained, supported and mentored by a clinician, such as a vestibular neurologist, who is also capable of managing complex vestibular presentations, allowing the provision of a comprehensive aTBI neurological service.

 

National and/or local guidelines delineating which cases would benefit from this service would improve management within the aTBI cohort. Useful strategies may additionally include formation of a therapy-led group from UK major trauma centres with the aim of developing peer support and sharing research, experiences and training. Such strategies have been found to be effective in other areas.[62] There is a precedent for physiotherapists taking on vestibular roles, however few occupational therapists appear to practice in this area in the UK.[20] This trend is not limited to TBI, as a majority of physiotherapists work with vestibular patients in other although mainly outpatient areas.[49 63] However, the roles physiotherapists and occupational therapists undertake can and do overlap, as noted by studies of inpatient stroke rehabilitation, where such overlap was found to benefit patient care.[64] Inpatient, multidisciplinary settings such as trauma centres, could therefore offer an environment where overlap of roles may provide additional staff capacity and support (a highlighted concern of respondents in our study). Acknowledging any reservations and working with all healthcare professionals involved would be key to improving future care pathways. Whether changes in behaviour would improve patient and service pathways remains unclear, however evidence in other clinical settings suggests therapist-led treatment can reduce referrals, hospital visits,[65] and patients' falls risk.[66 67]

Successful behaviour change requires a validated and well-designed intervention,[68 69] preferably designed using a framework to ensure relevant factors are considered.[68] Using the Behaviour change wheel,[70] suggestions for appropriate intervention functions and associated policies are shown in table 2.

## Limitations

Although this was a multicentre study, the two participating sites were from a similar locality, and thus the generalisability of our findings may be limited. While the participants sampled had a range of experience, we recruited only therapists and trauma and surgical doctors. Specialist brain injury nurses could have also been sampled, however previous literature notes therapists and doctors are most commonly involved in managing ward patients with vestibular dysfunction.[49] Notably, the involvement of neurologists with vestibular expertise in assessing aTBI—as occurs at our Trust—is uncommon in the UK. Finally, we did not ask participants to define dizziness at the outset of interviews. This may have resulted in slight subjective differences in participants' meanings of the term dizziness and should be considered when interpreting findings.

## CONCLUSIONS

This multicentre qualitative study highlights the barriers and facilitators to providing timely and accurate care for patients with aTBI with vestibular dysfunction. A range of role, knowledge and practical barriers to managing

**Table 2** Possible behaviour change interventions for therapists

| Essential condition | Intervention | Policies | BPPV-specific strategy |
|---|---|---|---|
| Capability | Education | Communication | Providing examples of therapists managing BPPV in other clinical areas |
| | | Guidelines | National or local guidelines recommending vestibular assessment and treatment in aTBI |
| | Training | Communication | Regular teaching on theoretical and practical assessment and treatment techniques for BPPV |
| Opportunity | Enablement | Service provision | Ensuring therapy teams have sufficient clinical capacity, role models and managerial support |
| | | Regulation | Medical (consultant level) support for complex cases |
| | | | Senior therapists being seen to use skills and train junior therapists |
| | | | Making relevant teams and individuals aware of a change in practice and a change in role |
| | Modelling | Communication | Clear expectations of what the role would involve |
| | | Regulation/Guidelines | Embedding a pathway of care for patients with vestibular diagnoses in TBI into acute services |
| | | | Data showing which patients would benefit from assessment and treatment |
| Motivation | Persuasion | Communication | Using data to show patient and service level benefits of assessing and treating patients early |

aTBI, acute traumatic brain injury; BPPV, benign paroxysmal positional vertigo.

vestibular dysfunction in aTBI were noted. Within our sample, trauma therapists appeared most suited to incorporate new assessment and treatment behaviours for BPPV into routine practice, supported by an appropriately trained multidisciplinary team. Theory-based strategies for implementing interventions to change behaviour are proposed. Further work is required to establish whether such changes in behaviour would result in patient and/or service-level improvements.

**Acknowledgements** The research team would like to acknowledge all the staff members who so willingly gave their time to be interviewed.

**Contributors** All authors contributed to this work. RMS designed/conceptualised the study/collected data/analysed data and drafted the manuscript. RMS holds a MRes and is an NIHR Doctoral Fellow and a physiotherapist by background. CB designed/conceptualised the study/analysed data and revised the manuscript. VT designed and conceptualised the study and revised the manuscript. JM designed/conceptualised the study and revised the manuscript. BMS designed/conceptualised the study/revised the manuscript, supervised the study, helped obtain funding for the study and accepts full responsibility for the work as the guarantor.

**Funding** Rebecca Smith is funded by the National Institute of Health Research [ICA-CDRF-2017-03-070]. The views expressed are those of the authors and not necessarily those of the NIHR or the Department of Health and Social Care. This work is also supported by grants to Barry Seemungal from the Medical Research Council (MRC) grant # MR/P006493/1, the Imperial Health Charity and the NIHR Imperial Biomedical Research Centre.

**Competing interests** RMS, CB, VT and JB declare no competing interests. BS is an ABN traumatic brain injury advisory committee member, a NICE guidelines review committee for head trauma and Editor in Chief for Journal of Concussion.

**Patient and public involvement** Patients and/or the public were involved in the design, or conduct, or reporting or dissemination plans of this research. Refer to the Methods section for further details.

**Patient consent for publication** Not applicable.

**Ethics approval** This study involves human participants and was approved by London Harrow Ethics Committee—17/LO/0434. Participants gave informed consent to participate in the study before taking part.

**Provenance and peer review** Not commissioned; externally peer reviewed.

**Data availability statement** Data are available upon reasonable request. Data are available on reasonable request.

**ORCID iDs**
Rebecca M Smith http://orcid.org/0000-0003-2628-9861
Barry M Seemungal http://orcid.org/0000-0002-6578-0904

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
