## [Reviewer comments · BMJ Open]

ARTICLE DETAILS

TITLE (PROVISIONAL)	Why are patients with acute traumatic brain injury not routinely assessed or treated for vestibular dysfunction in the UK? A qualitative study
AUTHORS	Smith, Rebecca; Burgess, Caroline; Tahtis, Vassilios; Marsden, Jonathan; Seemungal, Barry

VERSION 1 – REVIEW

REVIEWER	Whitney, Susan University of Pittsburgh, USA
REVIEW RETURNED	20-Sep-2022

GENERAL COMMENTS	This is a very well written and interesting paper. The condition of vestibular agnosia is much more common than previously described. My specific comments are included below: Page 3, line 25—it should be “were” not “data was” P. 4- replace Invisible with a better descriptor of what you mean. Page 4- line 31- define MDT Please add the original citation for the use of the framework approach: Ritchie J, Lewis J: Qualitative research practice: a guide for social science students and researchers. 2003, London: Sage p.9—line 23- what does outlying mean? I think that it critical to understand why you chose the 28 subjects? What is it that you were looking for? Page 8- I am not sure of the journal format but was surprised that notice of the IRB was at the end and not on line 47. I am seeing ethics approval typically in the text. Page 11- I think that getting stakeholder feedback from patients should have been in the methods. It just seemed to be dropped into the text after data analysis, which does not seem to be a good fit. It has nothing to do with data analysis. Figure 1-Can you be more specific in the title of the figure? Without reading the text, it does not make sense. Such as “five main themes” of what? Your questionnaire- I am very concerned about the prompts that you provided. It seems that you may have gotten responses that you wanted based on the prompts that you provided for the questions. Please provide more information in the text that supports that this is typical with the “framework” design that you utilized. Page 11 or 9--Why did you chose the 35 people that you asked? Why did you exclude the other 7 people who agreed to participate? Were those people any different than the 28 that you chose? It is confusing as you said that you invited 35- did ALL the people that your originally asked say yes?
--

Page 12- line 45—the A13 means nothing to me. It would be better to describe what that means here for it to be meaningful to a reader.

Page 12- line 56- what does identify patients and coordinate referrals mean? Do you mean identify vestibular disorders?

Page 12- please explain why you grouped the “therapists” together and if there were any differences between the 2 groups. On P.12- you do say that their responses were different so I don’t think that the groups can be combined.

P.12- Can you clarify what this mean “respondents also viewed their presence as a barrier to improving their vestibular knowledge and skills.”

P12- This sentence does not make sense. Please clarify.

Therapists appeared to devolve responsibility to patients, thought to be a sub-optimal process due to concerns patients would ‘fall through the net’.

p.12—Line 57 and 58- please clarify that this statement relates to vestibular care.

P.13- In all quotes, you need to describe rather than give a letter and number which is meaningless to anyone not from the UK about what A5 and others means.

p. 14. I would suggest that you take out all figures of speech such as ‘take their foot off the pedal’ unless it is a direct quote of a participant and if so, should be documented that way. Figures of speech are very challenging to understand internationally.

p. 15 I looked up “out of remit” on line 19 and still can’t make sense of its meaning. Can you rephrase it please? You also use the word remit on p.19—can you reword it please if possible.

p.23- it seems that your themes in Figure 2 and Figure 2 itself should have first been presented in your results section and then discussed here.

p.25—It seems odd to have a table in the discussion. I do like the table – it is very clear but it is unusual to be in the discussion.

What was the range of experience in their professional roles and their mean and range of experience working with persons with acute trauma for all the medical disciplines?

I would suggest that you add these references to support that managing the persons BPPV with aTBI may decrease their chance of falling as support for your training others to examine and treat uncomplicated cases of persons with aTBI with BPPV:

- Jumani K and Powell J, 2017 DOI: 10.1177/0003489417718847
- Gananca FF et al, Braz J Otorhinolayngol 2010

Elderly falls associated with benign paroxysmal positional vertigo. Ganança FF, Gazzola JM, Ganança CF, Caovilla HH, Ganança MM, Cruz OL. Braz J Otorhinolaryngol. 2010 Jan-Feb;76(1):113-20. doi: 10.1590/S1808-86942010000100019. PMID: 20339699

Table 1- please describe what a band 5-6 and, band 7-8, FY1-2 mean? I would strongly suggest that you look at the OT and the PT data separately unless you can convince me that they have the same training related to vestibular disorders. It was clear that they had different opinions as separate professions—you might want to highlight the differences that you identified.

	Why did you focus only on BPPV when vestibular hypofunction can occur also after aTBI? You could have asked about HINTS and how well they could determine if the person has vestibular hypofunction because of the aTBI. Supplementary data 3 chart Describe what an FY1 and SHO are. What does Ax Rx mean? What does OT B6, 7 and 8 mean? I am not sure that it is a good idea to identify the hospital (maybe hospital A and B?) as it might identify the person. What does unspecific identifier mean? What is CC? What does nonfunctional mean? What does theoretical versus non theoretical mean? What does role and remit mean? Why does PT, B5 have missing data?
--	---

REVIEWER	Gianoli, Gerard J Ear & Balance Institute
REVIEW RETURNED	24-Oct-2022

GENERAL COMMENTS	This study is a qualitative survey to investigate the reason why TBI patients are not routinely assessed for vestibular dysfunction. The authors surveyed 28 professionals at two major trauma centers. The results highlight the answer to the question posed. The first step to solving a problem is to understand what the problem is. The authors do an excellent job in this regard. While this is a qualitative report of subjective responses by the study's participants, I think it will be an invaluable addition to the literature upon which other authors may build. I have no major criticisms of this paper, although I may differ in their recommendations. That said, their recommendations are a great starting point if they are implemented.
--

VERSION 1 – AUTHOR RESPONSE

Response to reviewer's comments.

Thank you reviewing the manuscript and for all your comments and recommendations. We have addressed all comments and show changes below and where appropriate in the text (as noted by tracked changes).

Comment: Page 3, line 25—it should be “were” not “data was”

Response: Thank you for highlighting – this has been changed to “were”

Comment: P. 4- replace Invisible with a better descriptor of what you mean.

Response: Thank you – we have opted to keep the term in as it was used by respondents in the study to describe how they perceived dizziness. We have added quotation marks to indicate this.

Comment: Page 4- line 31- define MDT

Response: Thank you for highlighting – this has been changed to “multidisciplinary”

Comment: Please add the original citation for the use of the framework approach: Ritchie J, Lewis J: Qualitative research practice: a guide for social science students and researchers. 2003, London: Sage

Response: Thank you – this has been added

Comment: p.9—line 23- what does outlying mean?

Response: This term has been changed to “other”. The text now reads “Clinicians were invited by email to participate if (i) they routinely worked on a trauma, emergency, rehabilitation or other ward”.

Comment: I think that it critical to understand why you chose the 28 subjects? What is it that you were looking for?

Response: 35 clinicians were emailed an invitation to take part in the study. Of those 35, 7 declined to take part or did not reply and 28 clinicians provided their consent to take part in an interview. We sampled clinicians purposively – that is we selected clinicians based on their level of experience and their profession in order to gather a sample that encompassed a wide range of clinical experience (junior – senior) and different specialties (i.e. therapist/doctor). We ceased data collection when no new themes appeared to emerge from interviews (data saturation).

Comment: Page 8- I am not sure of the journal format but was surprised that notice of the IRB was at the end and not on line 47. I am seeing ethics approval typically in the text.

Response: The journal format states the ethics statement should precede the reference list

Comment: Page 11- I think that getting stakeholder feedback from patients should have been in the methods. It just seemed to be dropped into the text after data analysis, which does not seem to be a good fit. It has nothing to do with data analysis.

Response: Thank you. We have moved the stakeholder feedback from patients to the methods section.

Comment: Figure 1-Can you be more specific in the title of the figure? Without reading the text, it does not make sense. Such as “five main themes” of what?

Response: Thank you – this now reads “Figure 1. Five main themes relating to healthcare professionals experiences of managing vestibular dysfunction and their relation to the overarching concept of role”

Comment: Your questionnaire- I am very concerned about the prompts that you provided. It seems that you may have gotten responses that you wanted based on the prompts that you provided for the questions. Please provide more information in the text that supports that this is typical with the “framework” design that you utilized.

Response: Prompts and probes were utilised to stimulate discussion and were only used if the participant did not elaborate on their answer. This technique is commonly used within semi structured interviewing. We have provided more information in the text to make this clearer. The text now reads: “The topic guide included questions on how symptoms of vestibular dysfunction i.e. dizziness and imbalance were managed on the trauma ward. Prompts and probes were utilised to stimulate further discussion, if required [42]. This approach has also been used in other studies employing the TDF and Framework analysis [33,43]”

Comment: Page 11 or 9--Why did you chose the 35 people that you asked? Why did you exclude the other 7 people who agreed to participate? Were those people any different than the 28 that you chose? It is confusing as you said that you invited 35- did ALL the people that your originally asked say yes?

Response: Thank you. We have now made this clearer. The text now reads “Thirty-five healthcare professionals across two Major Trauma Centres were invited to take part: of those, seven declined to take part or did not reply to the invitation email and 28 participated”

Comment: Page 12- line 45—the A13 means nothing to me. It would be better to describe what that means here for it to be meaningful to a reader.

Response: We have amended the text to make this clearer. The text now reads “Quotes from participants are included to illustrate each of the five main themes. Quotes are followed by the pseudonym, profession and specialty of the participant”

Comment: Page 12- line 56- what does identify patients and coordinate referrals mean? Do you mean identify vestibular disorders?

Response: We have amended the text accordingly. It now reads: “Therapists (physiotherapists and occupational therapists) also expressed uncertainty about responsibility for managing vestibular dysfunction, although in practice they tended to identify patients with vestibular dysfunction and coordinate referrals”

Comment: Page 12- please explain why you grouped the “therapists” together and if there were any differences between the 2 groups. On P.12- you do say that their responses were different so I don’t think that the groups can be combined.

Response: There were similarities and differences between occupational therapists and physiotherapists in their responses to different questions. Where therapists are grouped together, this reflects similarity in their responses. Where there were differences, we have specifically linked themes or subthemes to a named profession. A defining feature of Framework analysis is the creation of a matrix which allows the researcher to compare cases (participants) by codes. During the analysis stage we were therefore able to compare and contrast different professions.

Comment: P.12- Can you clarify what this mean “respondents also viewed their presence as a barrier to improving their vestibular knowledge and skills.”

Response: We have reworded this. The text now reads: “Interestingly, although specialist input was universally noted to be positive, respondents also viewed the presence of specialists as a barrier to improving their own vestibular knowledge and skills”

Comment: P12- This sentence does not make sense. Please clarify. Therapists appeared to devolve responsibility to patients, thought to be a sub-optimal process due to concerns patients would ‘fall through the net’.

Response: We have reworded this. The text now reads: “Therapists appeared to devolve responsibility to patients to contact their General Practitioner if they had ongoing symptoms of vestibular dysfunction, thought to be a sub-optimal process due to concerns patients would ‘fall through the net’”

Comment: p.12—Line 57 and 58- please clarify that this statement relates to vestibular care.

Response: We have reworded this. The text now reads: “The uncertainty around responsibility and the variability noted in treatment pathways was thought to have a negative impact on vestibular care during and following admission”

Comment: P.13- In all quotes, you need to describe rather than give a letter and number which is meaningless to anyone not from the UK about what A5 and others means.

Response: We have made this clearer in the text. The text now reads: “Quotes from participants are included to illustrate each of the five main themes. Quotes are followed by the pseudonym, profession and specialty of the participant”

Comment: p. 14. I would suggest that you take out all figures of speech such as ‘take their foot off the pedal’ unless it is a direct quote of a participant and if so, should be documented that way. Figures of speech are very challenging to understand internationally.

Response: We have made this clearer in the text and noted where it is a direct quote from a participant. The text now reads: “As one participant described ‘we often take our foot off the pedal a little bit and dismiss other things as unimportant’.

Comment: p. 15 I looked up “out of remit” on line 19 and still can’t make sense of its meaning. Can you rephrase it please? You also use the word remit on p.19—can you reword it please if possible.

Response: We have reworded this. The text now reads: “Low confidence was noted in undertaking eye movement examinations, and interpretation of findings was felt to be out of the scope of their knowledge and skills”

Comment: p.23- it seems that your themes in Figure 2 and Figure 2 itself should have first been presented in your results section and then discussed here.

Response: This diagram was situated in the discussion as it uses the theme noted in the present study (that participants described dizziness as invisible) together with previous research noting that dizziness is not always a priority in aTBI and knowledge that bedside vestibular signs are not always easy to differentiate to a non-expert.

Comment: p.25—It seems odd to have a table in the discussion. I do like the table – it is very clear but it is unusual to be in the discussion.

Response: The table contains some proposed strategies to facilitate behaviour change. These strategies come directly from the results of the paper so we feel the discussion is an appropriate place for this.

Comment: What was the range of experience in their professional roles and their mean and range of experience working with persons with acute trauma for all the medical disciplines?

Response: Thank you. These details have been added. The text now reads: “In our sample, healthcare professionals had on average 47.4 months experience of working in trauma”

Comment: I would suggest that you add these references to support that managing the persons BPPV with aTBI may decrease their chance of falling as support for your training others to examine and treat uncomplicated cases of persons with aTBI with BPPV:

- Jumani K and Powell J, 2017 DOI: 10.1177/0003489417718847
- Gananca FF et al, Braz J Otorhinolaryngol 2010

Response: Thank you – these references have been added. The text now reads: “There is evidence in other settings that therapist led treatment can reduce referrals, hospital visits [65] and patients’ falls risk [66,67]”

Comment: Table 1- please describe what a band 5-6 and, band 7-8, FY1-2 mean? I would strongly suggest that you look at the OT and the PT data separately unless you can convince me that they have the same training related to vestibular disorders. It was clear that they had different opinions as separate professions—you might want to highlight the differences that you identified.

Response: We have removed Band 5-6 and Band 7-8 and FY1-2 from the table to make this clearer for an international audience. During analysis all participants’ stories (the themes and sub themes from their transcripts) were examined individually. The use of Framework analysis enabled us to

examine, contrast and compare data from Occupational therapists and Physiotherapists separately. Differences amongst therapists were highlighted in several different areas. E.g.

- Theme 1 (Current practice) The text reads “There was a sense amongst occupational therapists in particular, that managing vestibular dysfunction was outside of their remit and instead embedded in a physiotherapist’s role (due to their existing involvement in balance assessment)”
- Theme 3 – Assessment and treatment) The text reads “Where there was practical experience, this was limited to physiotherapists who were not routinely treating patients due to (i) low confidence secondary to limited patient exposure and insufficient training and mentorship, (ii) a reliance on visiting specialists, or (iii) the practicality of undertaking treatment in aTBI patients”
- Theme 4 (Practical barriers) The text reads “Whilst both physiotherapists and occupational therapists felt their previous vestibular training did not enable them to assess and treat independently, occupational therapists perceived their training was not comparable to that of physiotherapists. Although neither physiotherapists nor occupational therapists felt managing vestibular dysfunction was an expectation of their role, occupational therapists felt concerned that taking an interest would cross professional boundaries. Dizziness and imbalance were still perceived to be relevant to their role, however.”

There is no published data on the extent to which vestibular education is included amongst undergraduate physiotherapists or occupational therapists. This would be an interesting avenue for further focus and research.

Comment: Why did you focus only on BPPV when vestibular hypofunction can occur also after aTBI? You could have asked about HINTS and how well they could determine if the person has vestibular hypofunction because of the aTBI.

Response: This would be an interesting area for future focus. BPPV was a condition that came up amongst therapists as something they were partially familiar with; they had received previous theoretical and practical teaching on it but hadn’t ever been able to incorporate into practice. There were no therapists that had received teaching on identifying a vestibular hypofunction or being competent at a HINTS examination. Primarily the therapists interviewed are trauma therapists, not vestibular therapists and as such had varying degrees of previous vestibular education.

Comment: Supplementary data 3 chart

Describe what an FY1 and SHO are. What does Ax Rx mean?

What does OT B6, 7 and 8 mean?

I am not sure that it is a good idea to identify the hospital (maybe hospital A and B?) as it might identify the person. What does unspecific identifier mean?

What is CC? What does nonfunctional mean?

What does theoretical versus non theoretical mean?

What does role and remit mean?

Why does PT, B5 have missing data?

Response: Thank you. We have provided a key to the table and removed hospital identifiers. Please refer to amended file in Supplementary data.

VERSION 2 – REVIEW

REVIEWER	Whitney, Susan University of Pittsburgh, USA
REVIEW RETURNED	05-Dec-2022
GENERAL COMMENTS	The authors did a very good job with the revisions.